# Oral Antibiotics Alone versus Oral Antibiotics Combined with Mechanical Bowel Preparation for Elective Colorectal Surgery: A Propensity Score-Matching Re-Analysis of the iCral 2 and 3 Prospective Cohorts

**DOI:** 10.3390/antibiotics13030235

**Published:** 2024-03-03

**Authors:** Marco Catarci, Stefano Guadagni, Francesco Masedu, Massimo Sartelli, Leonardo Antonio Montemurro, Gian Luca Baiocchi, Giovanni Domenico Tebala, Felice Borghi, Pierluigi Marini, Marco Scatizzi

**Affiliations:** 1General Surgery Unit, Sandro Pertini Hospital, ASL Roma 2, 00157 Roma, Italy; leonardo.montemurro@aslroma2.it; 2General Surgery Unit, University of L’Aquila, 67100 L’Aquila, Italy; stefano.guadagni@univaq.it; 3Department of Applied Clinical Sciences and Biotechnology, University of L’Aquila, 67100 L’Aquila, Italy; francesco.masedu@univaq.it; 4General Surgery Unit, Santa Lucia Hospital, 62100 Macerata, Italy; massimosartelli@gmail.com; 5General Surgical Unit, Department of Clinical and Experimental Sciences, University of Brescia at the Azienda Socio Sanitaria Territoriale (ASST), 26100 Cremona, Italy; gianluca.baiocchi@unibs.it; 6Digestive & Emergency Surgery Unit, Santa Maria Hospital, 05100 Terni, Italy; gtebala@gmail.com; 7Oncologic Surgery Unit, Candiolo Cancer Institute, FPO-IRCCS, 10060 Candiolo, Italy; felice.borghi@ircc.it; 8General & Emergency Surgery Unit, San Camillo-Forlanini Hospital, 00152 Roma, Italy; marinipierluigi6@gmail.com; 9General Surgery Unit, Santa Maria Annunziata & Serristori Hospital, 50012 Firenze, Italy; marcoscatizzi60@gmail.com

**Keywords:** colorectal surgery, mechanical bowel preparation, oral antibiotics, anastomotic leakage, surgical site infections, morbidity

## Abstract

The evidence regarding the role of oral antibiotics alone (oA) or combined with mechanical bowel preparation (MoABP) for elective colorectal surgery remains controversial. A prospective database of 8359 colorectal resections gathered over a 32-month period from 78 Italian surgical units (the iCral 2 and 3 studies), reporting patient-, disease-, and procedure-related variables together with 60-day adverse events, was re-analyzed to identify a subgroup of 1013 cases (12.1%) that received either oA or MoABP. This dataset was analyzed using a 1:1 propensity score-matching model including 20 covariates. Two well-balanced groups of 243 patients each were obtained: group A (oA) and group B (MoABP). The primary endpoints were anastomotic leakage (AL) and surgical site infection (SSI) rates. Group A vs. group B showed a significantly higher AL risk [14 (5.8%) vs. 6 (2.5%) events; OR: 3.77; 95%CI: 1.22–11.67; *p* = 0.021], while no significant difference was recorded between the two groups regarding SSIs. These results strongly support the use of MoABP for elective colorectal resections.

## 1. Introduction

The earliest literature report on bowel decontamination and surgery dates back to 1899 [1]. During the last 80 years, the use of mechanical bowel preparation (MBP), oral antibiotics (oA), and perioperative intravenous antibiotic prophylaxis (PIVAP) to reduce the incidence of anastomotic leakage (AL) and surgical site infections (SSIs) in elective colorectal surgeries have shown time-related and geographic fluctuating trends, with clinical practice and guidelines remaining non-unique and inconclusive, despite the extraordinary number of published studies [2]. The use of MBP started at the beginning of the last century, becoming the usual practice in the 1930s until the beginning of the 1940s, when the use of multiple oral, non-absorbable sulfa derivatives, active only against aerobic species in the colon, was studied together with MBP [3]. After the Second World War, the discovery of several new oral, non-absorbable antibiotics active against aerobic and anaerobic species (aminoglycosides, tetracyclines, polimixines, macrolides, and, later on, nitroimidazoles) influenced bowel preparation before elective colorectal surgery, favoring oA combined with MBP (MoABP) and the diffusion of intraperitoneal resections with immediate anastomosis [4]. The landmark studies from surgeons in Chicago [5,6] using the oral administration of neomycin and erythromycin showed a dramatic reduction in AL and SSI rates, leading to the widespread diffusion of MoABP among North American surgeons, covering approximately 86% of cases at the end of the previous century [7]. At the same time, the introduction of parenteral cephalosporins and amoxicillin/clavulanate in the decades from the 1970s to 1980s shifted attention towards the major role of PIVAP in reducing SSI rates, and led to the current evidence [8] and the strong recommendation of the World Health Organization [9] for the administration of a single preoperative (30 to 120 min before the operation) intravenous dose of a cephalosporin and metronidazole, albeit with a conditional recommendation for the use of oA. At the beginning of the current century, several randomized controlled trials (RCTs) failed to demonstrate any clear benefit of MBP alone, supporting the concept of no bowel preparation (NBP), leading to the recommendation to avoid MBP in systematic reviews [10,11], in both the European [12] and Italian [13] Enhanced Recovery After Surgery (ERAS) society guidelines, and in the WHO guidelines [9]. Thereafter, the use of MoABP for colorectal surgery in North America dropped down to a 30–40% rate [14].

During the last ten years, however, the results of several large retrospective series stemming from the American College of Surgeons-National Surgical Quality Improvement Program (ACS-NSQIP) has led to the resurgence of the belief that MoABP significantly decreases SSIs and overall morbidity (OM) rates compared to NBP [15,16,17,18,19,20,21,22]. Consequently, the guidelines of four large North American societies (The American Society of Colon and Rectal Surgeons, the Society of American Gastrointestinal and Endoscopic Surgeons, the American Society for Enhanced Recovery, and the Perioperative Quality Initiative) recommended MoABP [23,24,25]. Thereafter, the number of patients treated with MoABP in North America rose again up to 80% of cases [26]. The rate of adoption of MoABP among European surgeons seems to have been more variable. It is currently used by 50% of Austrian–German [27] surgeons, while its use is much more limited (about 10% of cases) in Italy [28]. These figures will probably change in the near future, as very recently the European Association of Endoscopic Surgery, the European Society of ColoProctology, together with the Society of American Gastrointestinal and Endoscopic Surgeons, published a joint guideline recommending MoABP [29], albeit supported by low-quality evidence, due to the variable adherence to PIVAP and the great heterogeneity regarding oral antibiotics schedules [30].

During the last five years, three RCTs on this topic have been published. The first one, comparing NBP with MoABP [31], failed to detect significant differences in SSIs and AL rates, but it was largely underpowered. Another RCT, comparing NBP with oA [32], showed that the oral administration of ciprofloxacin 750 mg b.i.d. and metronidazole 250 mg t.i.d. the day before colon surgery significantly reduced SSIs. This trial, however, received some criticism related to the very low AL and major morbidity rates in both study arms [33]; its authors launched another RCT comparing oA with MoABP, which is currently still recruiting participants [34]. Finally, the third study compared PIVAP alone with PIVAP combined with oA [35], using different MBP schedules. It showed significantly reduced SSI rates in the oA arm, particularly when oA was coupled with MBP, although the PIVAP schedule did not include metronidazole, as has been recommended since 2014 [8]. Moreover, although several other RCTs have been launched, only one study comparing MoABP with MBP for rectal cancer [36] has completed the planned enrollment, although its results are not yet available. Unfortunately, an interesting, long-awaited, four-arm RCT comparing NBP with oA, MBP, and MoABP for colon resections [37] was recently closed before completion owing to poor accrual.

The great heterogeneity of both oral and intravenous antibiotic prophylaxis schedules, coupled with the heterogeneity of mechanical bowel preparations (polyethylene glycol, sodium phosphate, picosulfate, etc.), result in an extraordinary number of possible combinations potentially evaluable by RCTs. The current evidence regarding the “optimal” bowel preparation for elective colorectal surgery, therefore, is inconclusive because of the following (1) MoABP probably reduces SSIs as well as anastomotic leakage compared to MBP alone; (2) oA alone might be as effective as MoABP, but this cannot be clearly determined yet; and (3) whether NBP compared to MoABP has an influence on morbidity cannot be determined yet [2]. When conclusive evidence from RCTs is lacking, or when researchers need to assess treatment effects based on real-life data, a propensity score-matching analysis (PSMA) performed on data from prospective observational studies offers an alternative approach for estimating treatment effects. Based on these considerations, the Italian ColoRectal Anastomotic Leakage (iCral) study group estimated the effects of oA plus PIVAP (the treatment variable) versus MoABP plus PIVAP before elective colorectal surgery, through a PSMA of the data derived from two prospective, open-label, observational multicenter studies [38,39].

## 2. Materials and Methods

### 2.1. Study Design

This was a secondary, unplanned, ad hoc propensity score-matched re-analysis of two prospective cohorts of patients who had undergone colorectal surgery for malignant and benign diseases.

### 2.2. Patient Population and Data Collection

The iCral2 [38] and iCral3 [39] studies prospectively enrolled 8359 patients who underwent colorectal resection with anastomosis, according to explicit inclusion/exclusion criteria, in 78 surgical centers in Italy from January 2019 to September 2021. Both studies followed the Strengthening the Reporting of Observational Studies in Epidemiology (STROBE) guidelines [40].

The present PSMA study included 1013 (12.1%) patients selected from the parent studies according to explicit exclusion criteria (Figure 1) to control for data imbalance due to any treatment confounder. Most of the exclusions (81.8%) were based on a self-evident rationale (i.e., no bowel preparation, mechanical bowel preparation alone, no perioperative intravenous antibiotic prophylaxis, missing data regarding bowel preparation, perioperative steroids, mechanical bowel preparation different from PEG, and dialysis). The remaining criteria (neoadjuvant therapy, a proximal derivative stoma, urgency or delayed urgency, and an anastomosis within 6 cm from the external anal verge) accounted for 18.2% of the excluded cases, and were considered to limit the heterogeneity regarding one of the primary endpoints (anastomotic leakage). The descriptive variables considered for the 1013 patients are shown in Table 1. All 1013 patients were treated with PIVAP; however, the intravenous antibiotic schedules were not available.

The continuous variables were categorized according to their median values to reduce the number of unmatched cases. The true population of interest—oA—included 406 patients (40.0%); the control population—MoABP—included 607 patients (60.0%). Significant differences in age, nutritional status, indications of malignancy, type of surgical procedure, end-to-end anastomosis, hospital type, unit type, and the percentage of adherence to ERAS items were detected between the oA and MoA groups (Table 1). The patients in the MoABP control population prepared their bowels by drinking products containing polyethylene glycol the day before surgery. The patients in both groups received several different oral antibiotic schedules, the majority of which contained metronidazole, all of which provided both aerobic and anaerobic coverage (Table 2).

All the enrolled patients were followed up with for at least 8 weeks after surgery, recording and grading any adverse events [41,42], as well as any reoperations, readmissions, or death. Anastomotic leakage (AL) was defined according to the international consensus [43].

### 2.3. Outcomes

All the outcomes were calculated at 60 days after surgery. The primary outcomes were AL and SSIs, defined as superficial and/or deep surgical site infections (sdiSSIs), deep wound dehiscence, and/or abdominal collection/abscess [44]. The secondary outcomes were as follows: (1) overall morbidity (any adverse event), (2) major morbidity (any adverse event grade > II), and (3) reoperation (any unplanned operation) rates. In this retrospective study, mortality, sdiSSIs, deep wound dehiscence, and abdominal collection/abscess were not considered between the outcomes because the very small number of events in relation to the sample size (1013 patients) would make the statistical results of the comparison between the oA and MoABP groups burdened by inconsistency and unreliability [45,46].

### 2.4. Statistical Analysis

This was a retrospective PSMA of two prospective cohorts, with the sample sizes calculated and reported in the respective core papers [38,39]. The events per variable guideline were followed [45]. There were no missing data in the database for the 1013 patients. The target of the estimands was represented by the average treatment effect in the true population of interest (ATT). A propensity score-matching model [47,48] was used for the analysis (Figure 1). An adjusted logistic regression was used to estimate the propensity scores for the treatment and control groups. The exposure variable was a treatment that implied oA for the elective colorectal surgery. Twenty covariates potentially affecting the treatment [49] were selected: age, sex, American Society of Anesthesiologists (ASA) class, body mass index (BMI), diabetes, chronic renal failure, nutritional status measured through the Mini Nutritional Assessment—Short Form (MNA-SF) [50], surgery for malignancy, center volume, hospital type (academic/metropolitan versus local/regional), surgical unit type (general versus oncologic/colorectal), mini-invasive surgery, standard surgical procedure, operation length (minutes), intra- or extra-corporeal anastomosis, stapled versus handsewn anastomosis, end-to-end anastomosis, preoperative blood transfusion(s), intra- and/or postoperative blood transfusion(s), and overall ERAS pathway adherence rates. To ensure that the treatment groups were balanced [51], we performed the PSMA using the software “R©” (Version 4.2.2, The R Foundation© for Statistical Computing, Vienna, Austria, 2022). We used a nearest-neighbor approach with a logit distance metric and a caliper of 0.1 to minimize the differences between the groups. We also used an adjusted logistic regression to estimate the association between the treatment variable and outcomes. The balance of the matched groups was assessed by calculating the standardized mean difference (SMD) using a threshold of 0.1 (a standardized mean difference of less than 0.1 typically indicates a negligible difference between the means of the groups), and the general variance ratio (a variance ratio close to 1 indicates that variances are equal in the two groups). For outcome modeling, an adjusted logistic regression was performed based on a treatment variable represented by oA with elective colorectal surgery and on the same 20 covariates selected for the PSMA [52], which calculated the odds ratios (OR) and 95% confidence intervals (95%CI). The eventual effect of any unobserved confounder was tested through a sensitivity analysis [53] using the library “SensitivityR5” of the software R© (Version 4.2.2, The R Foundation^©^ for Statistical Computing, Vienna, Austria, 2022), which calculated the values (each 0.1 increment in the value represents a 10% odds of a differential assignment to treatment due to any unobserved variable). Sidak–Bonferroni’s adjustment for multiple comparisons was applied, setting α = 0.025, because the two primary outcomes were not independent and were selected based on the literature evidence [2].

## 3. Results

In this series of 1013 patients undergoing elective colorectal surgery for malignant and benign diseases, mortality events occurred in 4 patients (0.4%), 2 in the oA group and 2 in the MoABP group. Before propensity score matching, a univariate analysis of the entire population of 1013 patients showed no statistically significant differences in the primary and secondary outcomes between the oA and MoABP groups (Table 3).

After propensity score matching, 486 patients were included, and two groups of 243 patients each were generated (Figure 1): the oA group (the true population of interest) and the MoABP group (the control population). A good balance between the two groups was achieved (Figure 2 and Table 4), with a model variance ratio of 1.089.

After the multivariate logistic regression analysis for the endpoints for the 486 patients evaluated after score matching, oA versus MoABP was significantly associated with a higher risk of AL [14 (5.8%) vs. 6 (2.5%) events; OR: 3.77; 95%CI: 1.22–11.67; *p* = 0.021]. The sensitivity analysis calculated a Γ = 1 (*p* upper bound = 0.057). No difference was recorded between the two groups for SSIs [9 (3.7%) vs. 7 (2.9%) events; OR: 1.02; 95%CI: 0.31–3.29; *p* = 0.977]. The oA group was also significantly associated with a higher risk of major morbidity [25 (10.3%) vs. 9 (3.7%) events; OR: 4.55; 95%CI: 1.82–11.38; *p* = 0.001; Γ = 1.4; *p* upper bound = 0.038], and a higher risk of reoperation [16 (6.6%) vs. 5 (2.1%) events; OR: 5.05; 95%CI: 1.55–16.49; *p* = 0.007; Γ = 1.3; *p* upper bound = 0.037]. No significant differences were recorded between the two groups in terms of overall morbidity (Table 5).

According to the types of adverse events reported in the two groups (Table 6), the higher risk of major morbidity recorded in the oA group vs. MoABP group was significantly related to AL and superficial and/or deep surgical site infections (sdiSSIs).

## 4. Discussion

The effectiveness of oA and MoABP for reducing AL and SSI rates for elective colorectal resections remains largely controversial [2]. On the one hand, a well-designed RCT showed that oA alone is able to significantly reduce SSI rates compared to NBP, albeit with no influence on AL rates [32], while another large French RCT [33] showed the same finding, with the highest reduction achieved with MoABP, although the PIVAP schedule in this trial did not include metronidazole. On the other hand, two largely underpowered RCTs [54,55] showed inconclusive results. Analyses of the large retrospective databases of the ACS-NSQIP [17,18,20,21] and Veterans Affairs NSQIP [56] have suggested that both oA and MoABP may be equally effective in reducing AL and SSI rates compared to MBP alone or NBP. Therefore, while waiting for the results of the ongoing international RCT comparing oA to MoABP [34], it could be of particular interest to know how these different types of preoperative preparations work in real-life clinical practice.

To the best of our knowledge, the present study is the first PSMA to compare oA with PIVAP versus MoABP with PIVAP using the data derived from a prospective multicenter database, which represents a snapshot of the real-life clinical practice for 1013 Italian patients before elective colorectal surgery. There were no significant differences between the groups in terms of SSI rates, while there was a significantly higher risk of AL, MM, and reoperation in the group treated with oA (Table 5). The sensitivity analysis [53] showed Γ = 1 for AL, Γ = 1.4 for MM, and Γ = 1.3 for reoperation, meaning that 10%, 40%, and 30% of the patients in this study should have been treated with MoABP instead of oA, in order to alter the significant association between oA and the higher risk of AL, MM, and reoperation, respectively. The significantly higher MM risk in the oA group was significantly related to AL and sdiSSIs, among several other adverse events (Table 6), and the significantly lower risk of reoperation in the MoABP group may have been related to the causal link between AL and reoperation.

In summary, the results of this PSMA suggest that oA alone exposed the patients to a higher risk of AL and grade > II sdiSSIs. The reasons are mainly speculative, and rely on the conviction [57] that luminal feces may lead to the reduced efficiency of topically acting antibiotics. In 2016, Fry suggested that retained stool contains a large bulk of microbes, dietary fiber, and exfoliated cells that will not permit a reduction in the density of potential pathogens on the colonic mucosal surface with the use of oral antibiotics [58], supporting previous studies [3,5,59] performed in the 1940s and the 1970s. In patients treated with oA alone, it is possible that some members of the Bacteroidetes phylum [60] and other microbes, such as Enterococcus faecalis and Pseudomonas aeruginosa [57,61], can remain in the feces and colon mucosa and express enzymes that promote the degradation of synthesized tissue, leading to the vulnerability of the newly created anastomosis in response to the surgical trauma and resulting ischemia. The hypothesis that MBP could reduce the abundance of protective Bifidobacterium and Lactobacillus species, leading to higher rates of postoperative infections [62] appears less convincing, mainly because changes in the microbiota are only one of the factors that influence the rates of AL and postoperative complications after elective colorectal surgery [57,63].

Although the need for aerobic and anaerobic coverage is universally accepted, many different oral and intravenous antibiotic combinations have been previously reported [2], with prevalent geographic preferences. In the present study, many different antibiotics and administration schedules were used (Table 2), and because of the small number of AL events in each oral antibiotic and administration schedule subgroup, it was not possible to conclude which antibiotic and administration schedule is better for preventing AL. Over 100 trillion microorganisms (microbiota, including fungi, viruses, protozoans, and bacteria) are present in the gastrointestinal tracts of the hosts [60]. Approximately 80 bacterial species are present in the colorectal tract, differing between individuals according to many factors, including ethnicity, sex, age [60], cultural and social disparities [64], and antibiotic resistance due to extended-spectrum beta-lactamase (ESBL)-producing Enterobacterales due to previous antibiotic therapies [65]. Considering the recent shift in European guidelines towards recommending MoABP instead of NBP [23], many European surgeons (just like the authors) are currently asking themselves which oA regimen (molecules and schedules) should be implemented in their clinical practice. Based on the results of the most recent RCTs [32,35], a short-term (on the preoperative day) oral administration of a nitroimidazolic (i.e., ornidazole or metronidazole) combined with a quinolone (i.e., ciprofloxacin) is appealing, due to their optimal aerobic and anaerobic coverage. However, even a short-term course of oral metronidazole and ciprofloxacin [66] produces profound changes in the gut microbiota shortly after administration, with a drop in microbial diversity, an overgrowth of the genera Streptococcus and Lactobacillus, and an early loss of anaerobic bacterial taxa with important roles in short-chain fatty acid metabolism (colonic butyrate-producing communities) that have been demonstrated to be of paramount importance for colorectal mucosal integrity and anastomotic healing in animal studies [64,67]. Moreover, these changes require several months to return back to the baseline [66], and quinolones may be involved in the worldwide increasing incidence of a plague of multidrug-resistant microorganisms [68,69,70]. A possible answer may come, in the near future, from the ongoing Human Microbiome Project [71], whose worldwide mapping will allow for perioperative microbiome manipulation through the targeted administration of antibiotics, probiotics, or symbiotics to restore the ideal bowel flora by selecting specific bowel strains rather than continuing to search for an impossible “one-size-fits-all” elimination of the intestinal microbiota.

### Strengths and Limitations

The strengths of this study are the large number of enrolled patients in a well-defined time-lapse study, representing a real-life snapshot of the surgical units performing colorectal resections in Italy, and its PSMA methodology. Following recommendations for the use of propensity score methods [72,73], a rigorous patients selection from the parent population and the reasoned inclusion of 20 conditioning variables were performed to limit data imbalances. Moreover, both a clear and restrictive balance algorithm, together with the evaluation of the treatment effects through an adjusted multiple regression model including the same 20 covariates used for matching, were used (Figure 1). Finally, Rosenbaum’s sensitivity analysis for unmeasured confounders was applied [53].

On the other hand, this study has several limitations, and its results should be interpreted with caution: (a) a moderate heterogeneity of oral and intravenous antibiotic prophylaxis schedules, as reported by previous cohort studies [18,21,56]; (b) the exclusion criteria applied to the parent database (Figure 1) practically excluded any resection performed for low rectal cancers, making the results not applicable to this subgroup of patients; (c) several aspects of the health-acquired infections preventive bundle (preoperative whole-body bathing, hair removal, and skin decontamination) and each surgeon’s experience [74] were not measured in the parent studies; and (d) finally, further bias from residual unknown factors and potential measurement errors by the participating investigators may have had an impact on the results.

## 5. Conclusions

The present study contains an important warning, reporting that oA alone compared to MoABP before elective colorectal surgery was significantly associated with a higher risk of AL, MM, and reoperation. 

Future clinical research should be aimed at tailoring the administration of oral antibiotics, probiotics, and symbiotics according to the individual’s microbiome, instead of trying to adapt a “one size fits all” strategy of bowel preparation for elective colorectal surgery.

## Figures and Tables

**Figure 1 antibiotics-13-00235-f001:**
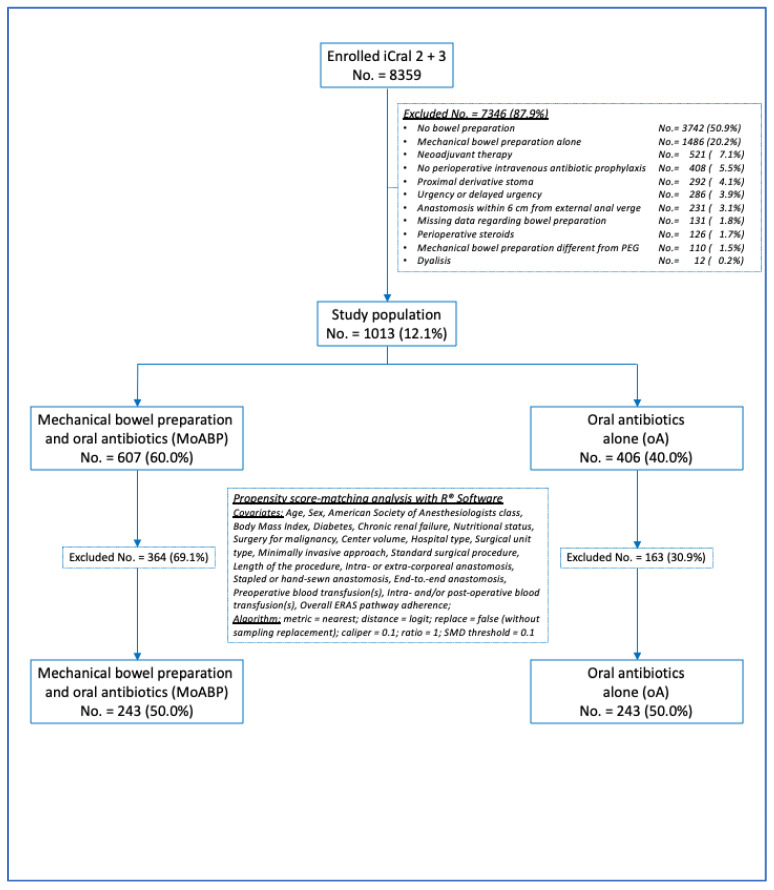
Study flowchart.

**Figure 2 antibiotics-13-00235-f002:**
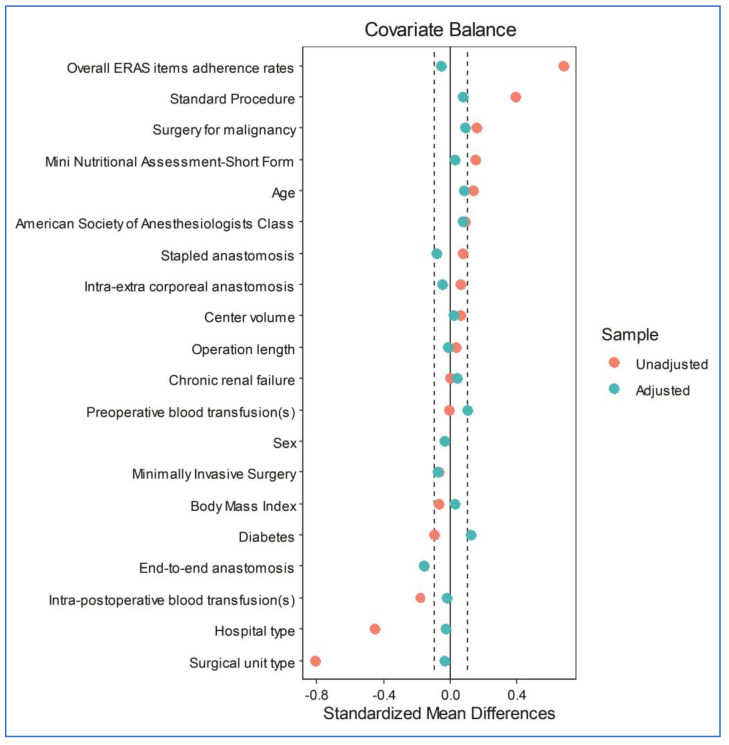
A Love plot of the covariates’ standardized mean differences between the treatment and control groups before and after matching; the vertical lines represent an interval of ± 0.1 within which the balance is considered acceptable.

**Table 1 antibiotics-13-00235-t001:** The descriptive analysis of the variables considered in the entire population.

			Overall		MoABP		oA		
			No. 1013		No. 607		No. 406		
Variables		Pattern	No.	%	No.	%	No.	%	* *p*
Age (years)		≤69	513	50.6	324	53.4	189	46.5	0.033
		>69	500	49.4	283	46.6	217	53.5	
Sex		Male	532	52.5	323	53.2	209	51.5	0.588
		Female	481	47.5	284	46.8	197	48.5	
ASA class		I–II	662	65.3	407	67.0	255	62.8	0.164
		III	351	34.7	200	33.0	151	37.2	
Body mass index (kg/m^2^)		≤24.67	507	50.1	295	48.6	212	52.2	0.259
		>24.67	506	49.9	312	51.4	194	47.8	
Diabetes		Yes	123	12.1	81	13.3	42	10.3	0.152
		No	890	87.9	526	86.7	364	89.7	
Chronic renal failure		Yes	45	4.4	27	4.5	18	4.4	0.991
		No	968	95.6	580	95.5	388	95.6	
MNA-SF		≤13	693	68.4	433	71.3	260	64.0	0.014
		>13	320	31.6	174	28.7	146	36.0	
Malignancy		Yes	739	73.0	427	70.4	312	76.9	0.022
		No	274	27.0	180	29.6	94	23.1	
	Diverticular disease		167	60.9	107	59.4	60	63.8	
	Endometriosis		2	0.8	0	0	2	2.2	
	Polyps		35	12.8	17	9.5	18	19.1	
	IBD		28	10.2	22	12.2	6	6.4	
	Other		42	15.3	34	18.9	8	8.5	
Mini-invasive surgery		No	113	11.1	62	10.2	51	12.6	0.245
		Yes	900	88.9	545	89.8	355	87.4	
	Laparoscopic		826	81.5	509	93.4	317	89.3	
	Robotic		32	3.2	17	3.1	15	4.2	
	Converted		42	4.2	19	3.5	23	6.5	
Standard procedure		Yes	859	84.8	488	80.4	371	91.4	0.000
	Right colectomy		407	47.4	199	40.8	208	56.1	
	Left colectomy		356	41.4	223	45.7	133	35.9	
	Anterior resection		96	11.2	66	13.5	30	8.1	
		No	154	15.2	119	19.6	35	8.6	
	Transverse colectomy		28	18.2	18	15.1	10	28.6	
	Splenic flexure colectomy		26	16.9	14	11.8	12	34.3	
	Hartmann reversal		16	10.4	12	10.1	4	11.4	
	(Sub)total colectomy		23	14.9	19	16.0	4	11.4	
	Other		61	39.6	56	47.0	5	14.3	
Anastomosis 1		Intracorporeal	732	72.3	432	71.2	300	73.4	0.343
		Extracorporeal	281	27.7	175	28.8	106	26.1	
Anastomosis 2		Stapled	868	85.7	514	84.7	354	87.2	0.263
		Handsewn	145	14.3	93	15.3	52	12.8	
Anastomosis 3		End to end	457	45.1	293	48.3	164	40.4	0.014
		Other shape	556	54.9	314	51.7	242	59.6	
Operation length		≤160′	521	51.4	316	52.1	205	50.5	0.625
		˃160′	492	48.6	291	47.9	201	49.5	
Hospital type		Met./Ac.	773	76.3	516	85.0	257	63.3	0.000
		Local/Regional	240	23.7	91	15.0	149	36.7	
Unit type		Colorectal/Oncologic	166	16.4	144	23.7	22	5.4	0.000
		General	847	83.6	463	76.3	384	94.6	
Center volume		<4 cases/month	257	35.2	221	36.4	136	33.5	0.342
		≥4 cases/month	656	64.8	386	63.6	270	66.5	
Preoperative BT(s)		Yes	43	4.2	26	4.3	17	4.2	0.941
		No	970	95.8	581	95.7	389	95.8	
Intra/postoperative BT(s)		Yes	58	5.7	43	7.1	15	3.7	0.023
		No	955	94.3	564	92.9	391	96.3	
ERAS adherence (%)		≤78.95	616	60.8	450	74.1	166	40.9	0.000
		˃78.95	397	39.2	157	25.9	240	59.1	
	Nutritional screening		711	70.2	410	67.6	301	74.1	
	Prehabilitation		411	40.6	183	30.2	228	56.2	
	Counseling		747	73.7	471	77.6	276	68.0	
	Immune enhancing nutrition		330	32.6	113	18.6	217	53.5	
	Antithrombotic prophylaxis		938	92.6	550	90.6	388	95.6	
	Preoperative carbohydrates load		582	57.5	326	53.7	256	63.1	
	No preanesthesia		741	73.2	448	73.8	293	72.2	
	Standard anesthesia protocol		980	96.7	584	96.2	396	97.5	
	Normothermia		974	96.2	576	94.9	398	98.0	
	Goal-directed or restrictive fluid therapy		898	88.7	539	88.8	359	88.4	
	Postoperative nausea/vomit prophylaxis		935	92.3	543	89.5	392	96.6	
	Multimodal analgesia		975	96.3	573	94.4	402	99.0	
	No nasogastric tube		882	87.1	491	80.9	391	96.3	
	Minimally invasive surgery		900	88.9	545	89.8	355	87.4	
	No drains		420	41.5	178	29.3	242	59.6	
	Urinary catheter <24–48 h		864	85.3	484	79.7	380	93.6	
	Early mobilization		842	83.1	469	77.3	373	91.9	
	Early oral feeding		726	71.7	374	61.6	352	86.7	
	Pre-discharge check		848	83.7	503	82.9	345	85.0	

MoABP: Mechanical bowel preparation plus oral antibiotics; oA: oral antibiotics; *: chi-square independence test with one degree of freedom; ASA: American Society of Anesthesiologists; MNA-SF: Mini Nutritional Assessment—Short Form; IBD: inflammatory bowel disease; Intracorporeal: anastomosis performed under visual control through a scope; Extracorporeal: anastomosis performed under direct visual control through an open access; Met./Ac.: metropolitan/academic; BT: blood transfusion(s); ERAS: enhanced recovery after surgery pathway.

**Table 2 antibiotics-13-00235-t002:** Oral antibiotic schedules in oA and MoA groups before propensity score matching.

Oral Antibiotic(s)	Administration Schedule	oA (No. 406)		MoABP (No. 607)		* *p*
		No.	%	No.	%	
Metronidazole (500 mg) Paromomycin (250 mg)	Started 2 days preop., TID Started 2 days preop., BID	118	29.1	29	4.8	0.006
Metronidazole (500 mg) Cefazolin (2000 mg)	Started 1 day preop., TID Started 1 day preop., OD	76	18.7	50	8.2	0.102
Metronidazole (500 mg) Trimethoprim (160 mg) + Sulfamethoxazole (800 mg)	Started 1 day preop., TID Started 1 day preop., TID	68	16.7	61	10.0	0.267
Metronidazole (500 mg) Neomycin + Bacitracin (300 mg)	Started 1 day preop., TID Started 1 day preop., TID	47	11.6	6	0.9	0.419
Metronidazole (500 mg) Amoxicillin (1000 mg)	Started 3 days preop., BID Started 3 days preop., BID	25	6.2	5	0.8	0.623
Metronidazole (250 mg) Ciprofloxacin (500 mg)	Started 1 day preop., TID Started 1 day preop., BID	20	4.9	21	3.5	0.823
Metronidazole (500 mg) Rifaximin (400 mg)	Started 7 days preop., TID Started 7 days preop., BID	5	1.2	9	1.5	0.963
Metronidazole (250 mg) Amoxicillin (1000 mg)	Started 1 day preop., BID Started 1 day preop., BID	0	0	50	8.2	n.e.
Paromomycin (250 mg)	Started 4 days preop., QID	44	10.8	0	0	n.e.
Paromomycin (1000 mg)	Started 1 day preop., OD	0	0	37	6.1	n.e.
Metronidazole (250 mg) Rifaximin (200 mg)	Started 1 day preop., TID Started 1 day preop., BID	3	0.8	0	0	n.e.
Metronidazole (500 mg) Rifaximin (200 mg)	Started 1 day preop., BID Started 1 day preop., BID	0	0	68	11.2	n.e.
Metronidazole (1000 mg) Rifaximin (400 mg)	Started 1 day preop., TID Started 1 day preop., TID	0	0	11	1.8	n.e.
Metronidazole (500 mg) Paromomycin (500 mg) Rifaximin (400 mg)	Started 1 day preop., BID Started 1 day preop., BID Started 1 day preop., BID	0	0	126	20.8	n.e.
Rifaximin (400 mg)	Started 1 day preop., TID	0	0	102	16.8	n.e.
Amoxicillin (1000 mg)	Started 3 days preop., TID	0	0	17	2.8	n.e.
Neomycin + Bacitracin (300 mg)	Started 1 day preop., TID	0	0	15	2.5	n.e.

oA: oral antibiotics; MoABP: mechanical bowel preparation plus oral antibiotics; *: t test for proportions comparison; OD: once daily; BID: 2 times per day; TID: 3 times per day; QID: 4 times per day; preop.: preoperatively; n.e.: test not executable because there are cells with insufficient values.

**Table 3 antibiotics-13-00235-t003:** The univariate analysis of outcomes in the entire population.

	Overall (No. 1013)		MoABP (No. 607)		oA (No.406)		
	No.	%	No.	%	No.	%	* OR (95%CI)
AL	37	3.7	21	3.5	16	3.9	1.14 (0.59–2.22), *p* = 0.689
SSIs	32	3.2	17	2.8	15	3.7	1.33 (0.66–2.70), *p* = 0.425
OM	239	23.6	135	22.2	104	25.6	1.20 (0.90–1.62), *p* = 0.215
MM	61	6.0	30	4.9	31	7.6	1.59 (0.95–2.67), *p* = 0.077
Reoperation	49	4.8	27	4.5	22	5.4	1.23 (0.69–2.19), *p* = 0.480

MoABP: mechanical bowel preparation plus oral antibiotics; oA: oral antibiotics; *: univariate odds ratio estimation with Wolf valuation of the confidence intervals (CIs); AL: anastomotic leakage; SSIs: superficial surgical site infections; OM: overall morbidity; MM: major morbidity.

**Table 4 antibiotics-13-00235-t004:** Variables’ distribution in control and treatment groups before and after propensity score matching.

		Before PSM				After PSM			
		MoABP No. 609	oA No. 406			MoABP No. 243	oA No. 243		
Covariates	Pattern			* *p*	** SMD			* *p*	** SMD
*Age*	≤69 *years*	324	283	0.039	0.14	128	118	0.414	0.08
	>69 *years*	189	217	0.039	−0.14	115	125	0.414	−0.08
*Sex*	*Male*	323	209	0.633	0.03	129	125	0.785	0.03
	*Female*	284	197	0.633	−0.03	114	118	0.785	−0.03
*ASA class*	*I–II*	407	255	0.186	0.09	165	156	0.444	0.08
	*III*	200	151	0.186	−0.09	78	87	0.444	−0.08
*Body mass index*	≤24.67 *Kg/m^2^*	295	212	0.287	−0.07	124	121	0.856	0.02
	>24.67 *Kg/m^2^*	312	194	0.287	0.07	119	122	0.856	−0.02
*Diabetes*	*Yes*	81	42	0.182	0.09	21	30	0.236	−0.12
	*No*	526	364	0.182	−0.09	222	213	0.236	0.12
*Chronic renal failure*	*Yes*	27	18	1.00	0.00	10	12	0.827	−0.04
	*No*	580	388	1.00	−0.00	233	231	0.827	0.04
*MNA-SF*	*≤13*	433	260	0.017	0.16	165	162	0.847	0.03
	*>13*	174	146	0.017	−0.16	78	81	0.847	−0.03
*Malignancy*	*Yes*	427	312	0.027	−0.15	167	176	0.426	−0.08
	*No*	180	94	0.027	0.15	76	67	0.426	0.08
*Mini-invasive surgery*	*Yes*	545	355	0.288	0.07	221	215	0.455	0.08
	*No*	62	51	0.288	−0.07	22	28	0.455	−0.08
*Standard procedures*	*Yes*	488	371	0.000	−0.32	208	213	0.594	−0.06
	*No*	119	35	0.000	0.32	35	30	0.594	0.06
*Anastomosis 1*	*Intracorporeal*	432	300	0.381	−0.06	177	172	0.687	0.05
	*Extracorporeal*	175	106	0.381	0.06	66	71	0.687	−0.05
*Anastomosis 2*	*Stapled*	514	354	0.304	−0.07	212	205	0.436	0.08
	*Handsewn*	93	52	0.304	0.07	31	38	0.436	−0.08
*Anastomosis 3*	*End to end*	293	164	0.016	0.16	116	97	0.010	0.16
	*Other shape*	314	242	0.016	−0.16	127	146	0.010	−0.16
*Operation length*	*≤* *160′*	291	201	0.671	−0.03	131	133	0.927	−0.02
	*˃* *160′*	316	205	0.671	0.03	112	110	0.927	0.02
*Hospital type*	*Met/Ac*	516	257	0.000	0.51	178	175	0.839	0.03
	*Local/Regional*	91	149	0.000	−0.51	65	68	0.839	−0.03
*Unit type*	*Col/Onc*	144	22	0.000	0.54	24	22	0.877	0.03
	*General*	463	384	0.000	−0.54	219	221	0.877	−0.03
*Center volume*	*Low*	221	136	0.377	0.06	65	63	0.918	0.02
	*High*	386	270	0.377	−0.06	178	180	0.918	−0.02
*Preoperative BT(s)*	*Yes*	26	17	1.00	0.00	8	13	0.372	−0.10
	*No*	581	389	1.00	−0.00	235	230	0.372	0.10
*Intra/Post-operative BT(s)*	*Yes*	43	15	0.033	0.15	15	14	1.00	0.02
	*No*	564	391	0.033	−0.15	228	229	1.00	−0.02
*ERAS adherence*	*≤* *78.95%*	450	166	0.000	0.71	140	147	0.580	−0.06
	*˃78.95%*	157	240	0.000	−0.71	103	96	0.580	0.06

MoABP: mechanical bowel preparation plus oral antibiotics; oA: oral antibiotics; *: Student’s test for proportions; **: standardized mean difference; ASA: American Society of Anesthesiologists; MNA-SF: Mini Nutritional Assessment—Short Form; Intracorporeal: anastomosis performed under visual control through a scope; Extracorporeal: anastomosis performed under direct visual control through an open access; Met/Ac: metropolitan/academic; Col/Onc: colorectal/oncologic; BT(s): blood transfusion(s); ERAS: enhanced recovery after surgery.

**Table 5 antibiotics-13-00235-t005:** The multivariate logistic regression analysis of the endpoints considered for the 486 patients evaluated using the PSMA.

					Propensity Score-Matched Analysis			
	MoABP No. 243		oA No. 243				* Sensitivity	
Endpoint	No.	%	No.	%	OR (95%CI)	*p*	Γ	** *p*
Anastomotic leakage	6	2.5	14	5.8	3.77 (1.22–11.67)	0.021	1.0	0.057
SSIs	7	2.9	9	3.7	1.02 (0.31–3.29)	0.977		
Overall morbidity	49	20.2	64	26.3	1.52 (0.96–3.40)	0.075		
Major morbidity	9	3.7	25	10.3	4.55 (1.82–11.38)	0.001	1.4	0.038
Reoperation	5	2.1	16	6.6	5.05 (1.55–16.49)	0.007	1.3	0.037

MoABP: mechanical bowel preparation plus oral antibiotics; oA: oral antibiotics; *: Rosenbaum’s sensitivity analysis; **: *p* upper bound; OR (95%CI): odds ratio estimation with 95% confidence intervals; SSIs: surgical site infections.

**Table 6 antibiotics-13-00235-t006:** Adverse events contributing to overall morbidity and major morbidity in 486 patients evaluated using PSMA.

	MoABP No. 243		oA No. 243			
Adverse Events	OM (%)	MM (%)	OM (%)	MM (%)	* *p* (OM)	* *p* (MM)
Anastomotic leakage	6 (2.5)	4 (1.6)	14 (5.8)	12 (4.9)	0.068	0.042
sdiSSIs	2 (0.8)	0 (0)	6 (2.5)	4 (1.6)	0.154	0.045
Deep wound dehiscence	1 (0.4)	1 (0.4)	2 (0.8)	2 (0.8)	0.562	0.562
Abdominal collection/abscess	4 (1.7)	1 (0.4)	3 (1.2)	3 (1.2)	0.703	0.315
Small bowel obstruction	7 (2.9)	5 (2.1)	4 (1.6)	3 (1.2)	0.360	0.476
Anastomotic bleeding	2 (0.8)	1 (0.4)	8 (3.3)	1 (0.4)	0.055	1.00
Abdominal bleeding	2 (0.8)	1 (0.4)	1 (0.4)	1 (0.4)	0.562	1.00
Small bowel perforation	0 (0)	0 (0)	0 (0)	0 (0)	n.e.	n.e.
Trocar/wound site bleeding	1 (0.4)	0 (0)	1 (0.4)	0 (0)	1.00	n.e.
Anemia	6 (2.5)	0 (0)	9 (3.7)	1 (0.4)	0.431	0.317
Paralytic ileus	9 (3.7)	0 (0)	8 (3.3)	0 (0)	0.805	n.e.
Fever	6 (2.5)	0 (0)	8 (3.3)	0 (0)	0.588	n.e.
DVT/PE	0 (0)	0 (0)	1 (0.4)	0 (0)	0.317	n.e.
Neurologic	1 (0.4)	1 (0.4)	1 (0.4)	0 (0)	1.00	0.317
Pneumonia and pulmonary failure	5 (2.1)	0 (0)	7 (2.9)	2 (0.8)	0.559	0.156
Urinary retention	1 (0.4)	0 (0)	2 (0.8)	0 (0)	0.562	n.e.
Urinary tract infection	0 (0)	0 (0)	1 (0.4)	0 (0)	0.317	n.e.
Acute renal failure	0 (0)	0 (0)	4 (1.6)	0 (0)	0.062	n.e.
Acute mesenteric ischemia	0 (0)	0 (0)	0 (0)	0 (0)	n.e.	n.e.
Acute peptic ulcer/erosive gastritis	0 (0)	0 (0)	0 (0)	0 (0)	n.e.	n.e.
Cardiac dysfunction and failure	2 (0.8)	1 (0.4)	2 (0.8)	2 (0.8)	1.00	0.562
Other	14 (5.8)	1 (0.4)	7 (2.9)	2 (0.8)	0.118	0.562

MoABP: mechanical bowel preparation plus oral antibiotics; oA: oral antibiotics; OM: overall morbidity (number of events); MM: major morbidity (number of events); *: chi-square independence test with one degree of freedom; sdiSSIs: superficial and/or deep incisional surgical site infections; DVT: deep venous thrombosis; PE: pulmonary embolism; n.e.: test not executable because there are cells with insufficient values.

## Data Availability

Individual participant-level anonymized datasets are available upon request by contacting the study coordinator.

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
