# Peer review of "Oral Antibiotics Alone versus Oral Antibiotics Combined with Mechanical Bowel Preparation for Elective Colorectal Surgery: A Propensity Score-Matching Re-Analysis of the iCral 2 and 3 Prospective Cohorts"

_antibiotics, 2024, doi:10.3390/antibiotics13030235_

Round 1

Reviewer 1 Report

Comments and Suggestions for Authors

Dear Authors,

Thank you for the opportunity to contribute to the evaluation of the manuscript titled "Oral antibiotics alone versus oral antibiotics combined with mechanical bowel preparation for elective colorectal surgery: a propensity score-matching re-analysis of the iCral2 and iCral3 prospective cohorts, by Catarci et al."

The research presents a retrospective analysis of data from a prospective study on colorectal resections, focusing on the role of oral antibiotics alone or combined with mechanical bowel preparation for colorectal surgery and their rates of adverse events.

The research raises a relevant issue and potentially original work for the field of colorectal surgery research. As a relevant point, I highlight the reanalysis of prospective data from a large multicenter database, applying propensity score matching, in addition to the sample size (1,013 patients), which is considerable, increasing the reliability of the results. The research presents a new major finding: the significantly higher risk of anastomotic leakage in the oA group compared to MoABP.

Regarding the methodology, two points caught my attention:

1- Large disparity in group sizes: The MoABP group being 50% larger than the oA group may affect the analysis and interpretation of the results.

2- Limited information on PSMA: The methodology mentions optimizing the effectiveness of PSMA but does not provide details on specific matching variables or algorithms used.

Therefore, I suggest that the authors justify the reason for including a larger MoABP group and its potential impact on the results, as well as provide more details about the PSMA methodology, including the specific variables used for matching and the chosen algorithm.

Finally, I thank you once again for the opportunity to contribute to such valuable and important work.

Author Response

  • Large disparity in group sizes: The MoABP group being 50% larger than the oA group may affect the analysis and interpretation of the results.

This is a very relevant issue in all real-life observational studies. Disparity in the two groups size is a matter of fact deriving from the parent studies. In our context we dealt with this issue using a PSM analysis balancing the two treatments for the observed heterogeneity according to the available 20 covariates. As a result, we achieved two groups of 243 patients, using a 1:1 matching with the caliper 0.1. This methodological choice is due to the need to account for bias as much as possible, while preserving an acceptable power of the study to detect differences in the outcomes. Potential unobserved heterogeneity has been addressed using a sensitivity analysis according to Rosenbaum's Gamma.

  • Limited information on PSMA: The methodology mentions optimizing the effectiveness of PSMA but does not provide details on specific matching variables or algorithms used.

We totally disagree with the reviewer. All the details regarding PSMA methodology are clearly and sheerly reported in the subsection “Statistical analysis”, as well as in in Fig. 1 and 2 and Tables 1 and 3, allowing the reproducibility of the analysis according to the EQUATOR guidelines for PSMA [ref. # 72 and 73].

  • Therefore, I suggest that the authors justify the reason for including a larger MoABP group and its potential impact on the results, as well as provide more details about the PSMA methodology, including the specific variables used for matching and the chosen algorithm.

Please refer to the answers to the previous points.

Reviewer 2 Report

Comments and Suggestions for Authors

1.       What is the novelty of the present study? Similar studies are present in the literature, https://bmjopen.bmj.com/content/11/7/e051269 . Please discuss and justify.

2.       What are the future implications of this study?

3.       Conclusions should be extended including future implications of this study.

4.       There are multiple limitations associated with this study. Please include a separate section describing the limitations of the present study. 

Author Response

  1. What is the novelty of the present study? Similar studies are present in the literature, https://bmjopen.bmj.com/content/11/7/e051269 . Please discuss and justify.

We thank the reviewer for this important point. However, as clearly described in the introduction and in the discussion, the issue regarding the role of oral antibiotics alone (oA) versus oral antibiotics coupled with mechanical bowel preparation (MoABP) is far from being solved, as previous RCTs on this specific topic [ref. #54, 55, cited at page 12, lines 376-377) gave conflicting results. Actually, the paper cited by the reviewer is just the protocol of a still ongoing RCT, comparing mechanical bowel preparation (MBP) alone versus MoABP. It is, therefore, not pertinent to the current analysis.

  1. What are the future implications of this study?

Future implications of this study are the need to 1) avoid the administration of oral antibiotics alone; 2) tailor the administration of oral antibiotics, probiotics and symbiotics according to the individual microbiome instead of continuing to search for a “one size fits all” strategy of bowel preparation for elective colorectal surgery.

  1. Conclusions should be extended including future implications of this study.

The above mentioned consideration has been now included in the conclusions.

  1. There are multiple limitations associated with this study. Please include a separate section describing the limitations of the present study. 

All the limitations of the study have been reported in the discussion. However, since the reviewer did not catch them, we listed them in a separate section at the end of the discussion.

Reviewer 3 Report

Comments and Suggestions for Authors

The authors should consider the followings:

"The effectiveness of oA and MoABP in reducing AL and SSIs rates for elective colo- rectal resections remains largely controversial" The authors should expand their discussion in the controversial points.

The authors added that "while waiting for the results of the ongoing international RCT comparing oA to MoABP [34]". Instead of reanalysis and pooling of previous study (without new clinical data), it is sometimes indeed essential to look into the perspective with a new international trials (if the research ethics justified appropriate).

The authors should justify whether the different number of patients per group would affect the analysis, "The true population of interest, defined by the treatment variable oA, included 406 patients (40.0%); the control population, defined by the variable MoABP, included 607 patients(60.0%)."

As in figure 1, the study excluded "7346 patients". The authors may further supplement rationales for each exclusion items, in the methodology section.

In table 1, did the authors consider the ethnicity of the patients? If yes, why is it not listed in table 1? If no, why not? And would this affect the stratification pipelines?

"All enrolled patients were followed up for at least 8 weeks after surgery" what were the frequency of follow-up? And why did the rationales of ending the follow up after 8 weeks?

"Informed Consent Statement: Informed consent was obtained from all subjects involved in the parent studies." The authors should look into the clauses whether the parent studies truly resolved the permissions of the current study.

"Data Availability Statement: Individual participant-level anonymized datasets are available upon reasonable request by contacting the study coordinator." Please define the "reasonable" and give examples. 

The authors should list the limitations of the current study.

The authors should not overstate (nor overclaim) the results in the conclusion part.

There are a broad types of different antibiotics with varied course of adminstration (i.e. as shown in Table 2, Oral antibiotic schedules in the oA and MoA groups) The authors may not overly generalized the use or not use of antibiotics as a whole, but to consider the individual types and courses of the antibiotics treatments.

Comments on the Quality of English Language

Moderate editing in English language is needed.

Author Response

  • "The effectiveness of oA and MoABP in reducing AL and SSIs rates for elective colorectal resections remains largely controversial" The authors should expand their discussion in the controversial points.

A clear description of all the controversial points on the topic was already given both in the introduction and in the discussion. The authors, therefore, do not see where and how the discussion could be expanded.

  • The authors added that "while waiting for the results of the ongoing international RCT comparing oA to MoABP [34]". Instead of reanalysis and pooling of previous study (without new clinical data), it is sometimes indeed essential to look into the perspective with a new international trials (if the research ethics justified appropriate).

The reviewer is right. However, when evidence form RCTs is still lacking or controversial, PSMA of prospective data (although not new) may, as in the present study, give useful hints to clinical practice (i.e. avoiding the administration of oral antibiotics alone). Moreover, the results of the ongoing international RCT comparing oA to MoABP [ref. 34] will not be available before the end of year 2026.

  • The authors should justify whether the different number of patients per group would affect the analysis, "The true population of interest, defined by the treatment variable oA, included 406 patients (40.0%); the control population, defined by the variable MoABP, included 607 patients(60.0%)."

This is a very relevant issue in all real-life observational studies. Disparity in the two groups size is a matter of fact deriving from the parent studies. In our context we dealt with this issue using a PSM analysis balancing the two treatments for the observed heterogeneity according to the available 20 covariates. As a result, we achieved two groups of 243 patients, using a 1:1 matching with the caliper 0.1. This methodological choice is due to the need to account for bias as much as possible, while preserving an acceptable power of the study to detect differences in the outcomes. Potential unobserved heterogeneity has been addressed using a sensitivity analysis according to Rosenbaum's Gamma.

  • As in figure 1, the study excluded "7346 patients". The authors may further supplement rationales for each exclusion items, in the methodology section.

All the exclusion criteria are detailed in the study flowchart (Fig. 1), and their rationales were not listed in the methodology section to avoid significant lengthening of the paper. However, most of the exclusions (81.8%) have a self-evident rationale (i.e.: No bowel preparation, Mechanical bowel preparation alone, No perioperative intravenous antibiotic prophylaxis, Missing data regarding bowel preparation, Perioperative steroids, Mechanical bowel preparation different from PEG, Dyalisis). The remaining criteria (Neoadjuvant therapy, Proximal derivative stoma, Urgency or delayed urgency, Anastomosis within 6 cm from external anal verge), accounting for 18.2% of excluded cases, were accounted for to limit heterogeneity regarding one of the primary endpoints (anastomotic leakage). Anyway, these reasons were added in the methodology.

  • In table 1, did the authors consider the ethnicity of the patients? If yes, why is it not listed in table 1? If no, why not? And would this affect the stratification pipelines?

Thank you for rising this very good point; however, the near totality of patients in this Italian cohort is composed by caucasian patients. Therefore, ethnicity was not recorded in the parent studies and is not considered in the present analysis.

  • "All enrolled patients were followed up for at least 8 weeks after surgery" what were the frequency of follow-up? And why did the rationales of ending the follow up after 8 weeks?

The frequency of follow-up is described in the parent studies, and is not repeated here to avoid excessive lengthening of the manuscript. The rationale of ending follow up at 8 weeks lies in the need to record any adverse event at 60 days rather than at 30 days after surgery, since more than 30% of the adverse events considered as endpoints of this study was recorded beyond the usual follow-up of 30-day after surgery. After the time span of 8 weeks there simply is no reason to prolong follow-up, as the probability of the considered endpoints lies below 0.5% of cases.

  • "Informed Consent Statement: Informed consent was obtained from all subjects involved in the parent studies." The authors should look into the clauses whether the parent studies truly resolved the permissions of the current study.

The informed consent obtained by any participant in the parent studies included permission to perform any primary and secondary analysis of the results.

  • "Data Availability Statement: Individual participant-level anonymized datasets are available upon reasonable request by contacting the study coordinator." Please define the "reasonable" and give examples.

The reviewer is right, as “reasonable” is somehow subjective. Therefore, we changed it to “explicit”.

  • The authors should list the limitations of the current study.

All the limitations of the study have been reported in the discussion. However, since the reviewer did not catch them, we listed them in a separate section at the end of the discussion.

  • The authors should not overstate (nor overclaim) the results in the conclusion part.

Good point, we revised the conclusions accordingly.

  • There are a broad types of different antibiotics with varied course of adminstration (i.e. as shown in Table 2, Oral antibiotic schedules in the oA and MoA groups) The authors may not overly generalized the use or not use of antibiotics as a whole, but to consider the individual types and courses of the antibiotics treatments.

The reviewer is perfectly right, as the heterogeneity in both intravenous and oral antibiotics used for prophylaxis is the main reason for the inconclusive evidence gathered by previous RCTs to date. On the other hand, the limited number of the events in the endpoint considered in the present study according to the single oral antibiotic and administration schedule subgroup (Table 2) did not allow the analysis suggested by the reviewer, as clearly reported in the discussion (page 12, lines 411-416).

  • Comments on the Quality of English Language Moderate editing in English language is needed.

The entire manuscript was revised for grammar and syntax.

Round 2

Reviewer 2 Report

Comments and Suggestions for Authors

The authors successfully responded to the reviewer's comments and updated the manuscript as well.